# Hypertension Related to Obesity: Pathogenesis, Characteristics and Factors for Control

**DOI:** 10.3390/ijms232012305

**Published:** 2022-10-14

**Authors:** Paul El Meouchy, Mohamad Wahoud, Sabine Allam, Roy Chedid, Wissam Karam, Sabine Karam

**Affiliations:** 1Department of Internal Medicine, MedStar Health, Baltimore, MD 21218, USA; 2Department of Internal Medicine, Tufts Medical Center, Boston, MA 02111, USA; 3Faculty of Medicine and Medical Sciences, University of Balamand, El Koura P.O. Box 100, Lebanon; 4College of Osteopathic Medicine, William Carey University, Hattiesburg, MS 39401, USA; 5Department of Internal Medicine, University of Kansas School of Medicine, Wichita, KS 67214, USA; 6Division of Nephrology and Hypertension, University of Minnesota, Minneapolis, MN 55414, USA

**Keywords:** hypertension, obesity, pathogenesis, characteristics, genetics, treatment

## Abstract

The World Health Organization (WHO) refers to obesity as abnormal or excessive fat accumulation that presents a health risk. Obesity was first designated as a disease in 2012 and since then the cost and the burden of the disease have witnessed a worrisome increase. Obesity and hypertension are closely interrelated as abdominal obesity interferes with the endocrine and immune systems and carries a greater risk for insulin resistance, diabetes, hypertension, and cardiovascular disease. Many factors are at the interplay between obesity and hypertension. They include hemodynamic alterations, oxidative stress, renal injury, hyperinsulinemia, and insulin resistance, sleep apnea syndrome and the leptin-melanocortin pathway. Genetics, epigenetics, and mitochondrial factors also play a major role. The measurement of blood pressure in obese patients requires an adapted cuff and the search for other secondary causes is necessary at higher thresholds than the general population. Lifestyle modifications such as diet and exercise are often not enough to control obesity, and so far, bariatric surgery constitutes the most reliable method to achieve weight loss. Nonetheless, the emergence of new agents such as Semaglutide and Tirzepatide offers promising alternatives. Finally, several molecular pathways are actively being explored, and they should significantly extend the treatment options available.

## 1. Introduction

The World Health Organization (WHO) refers to obesity as abnormal or excessive fat accumulation that presents a risk to health [1]. The body mass index (BMI), which is calculated using weight and height as kg/m² [2,3], is used to screen for obesity in adults, with a cut-off set at 30 kg/m² in adults by the Centers for Disease Control and Prevention (CDC) [4]. People having a BMI of 30.0 to 34.9 are considered as having obesity class I, those with a BMI of 35 to 39.9 belong to class II obesity, and those with a BMI of 40 and above are considered as class III patients or as having severe obesity, formerly known as morbid obesity [4]. However, this tool has several limitations: it does not reflect the pathogenesis of the disease, does not permit distinguishing the contribution of fat versus muscle mass, and does not indicate the extent of the impact on health [5,6]. Abdominal obesity is a better reflection of risk and is usually assessed by measuring waist circumference [7].

The American Association of Clinical Endocrinologists (AACE) first designated the condition of obesity as a chronic disease in 2012 [8]. Since then, both the cost and the burden of disease attributed to obesity have experienced a worrisome increase. In 2016, the total cost accounted for 47% of that of chronic diseases [9]. Also, the 2012 obesity and severe obesity forecast through 2030 suggested that 51% of the population will be obese as of 2030, with a 130% increase in morbid obesity prevalence [10].

Obesity and hypertension are closely interrelated as abdominal obesity interferes with the endocrine and immune systems and carries a greater risk for insulin resistance, diabetes, hypertension, and cardiovascular disease [3,11,12]. Moreover, obesity is recognized as a major risk factor for hypertension in both adults and children, regardless of race, ethnicity, and sex [13,14]. Two landmark studies that investigated this association are the Nurse’s Health Study and the Framingham Heart Study [15,16]. The Nurse’s Health Study is a prospective cohort study of 83,882 adult women who were followed up for 16 years. The results showed that: (a) increased BMI was associated with the development of hypertension, (b) the relative risks for hypertension were 1.7 and 5.2 in women who gained 5–10 kg and >25 kg, respectively, and (c) 40% of the new-onset hypertension cases were attributed to overweight and obesity [15].

Similarly, the Framingham Heart Study, in which participants were followed up for up to 44 years, showed that increased adiposity accounted for 26% and 28% of cases of hypertension in men and women, respectively [16]. Moreover, the prevalence of hypertension increases across BMI levels with a prevalence of 87% in obesity class III [17]. Therefore, as obesity attains endemic proportions, hypertension risk and impact on health is only amenable to worsen. However, the association between adiposity and blood pressure is not straightforward, and many factors are at interplay between obesity and hypertension. In this review, we aim to detail some of the factors behind the pathogenesis of hypertension associated with obesity, along with the characteristics of hypertension in obese patients. We will also explore current and potential future pathways to eradicate obesity.

## 2. Pathogenesis of Hypertension Associated with Obesity

The factors behind obesity-induced hypertension are multiple and often take effect simultaneously. They include: hemodynamic alterations with changes in the generation of endothelium-derived constricting as well as relaxing factors, disruption of molecular signaling, increased oxidative stress, renal injury, hyperinsulinemia and insulin resistance, sleep apnea syndrome, and the leptin-melanocortin pathway [18,19]. Cardiovascular and hemodynamic alterations differ according to the distribution of obesity as patients with peripheral obesity have higher cardiac output (CO) and lower systemic vascular resistance (SVR), while patients with central obesity have lower CO and higher SVR [20].

At the cellular level, adipose tissue contributes to endothelial dysfunction by secreting multiple hormones and paracrine signals known as adipokines. These molecules play an important physiological role in regulating vascular tone. In the case of obesity, there is excessive secretion of pro-inflammatory and vasoactive adipokines such as angiotensinogen, angiotensin II, aldosterone, and resisting, along with an increase in plasma renin activity [21,22,23]. In addition, renin receptor expression is increased in human visceral adipose tissue [24]. Obesity also structurally alters the endothelial tissue. A notable example is the endothelial glycocalyx layer. Mice that were fed a strictly high-fat diet were found to have thinner and stiffer glycocalyx. This in turn, is associated with a significant decrease in the sensitivity of inwardly rectifying K^+^ (Kir) channels. These channels are responsible for flow-induced vasodilation [25].

Regarding oxidative stress, the metabolism of excess free fatty acids (FFAs) through β-oxidation and the TCA cycle produces excess reactive oxygen species (ROS) [26]. ROS generation is also increased in adipocyte cells as FFAs can stimulate NADPH oxidase, an enzyme involved in superoxide radical, nutrient-based ROS generation, and vascular injury [18]. Further oxidative stress is caused by FFAs, which are released from over-accumulated fat and can activate NADPH oxidase indirectly by stimulating the production of diacylglycerol, which activates protein kinase C, a direct activator of NADPH [27]. In turn, activation of NADPH oxidase, may contribute to the progression of hypertension through activation of the central sympathetic system [28]. The hyperlipidemic state of obesity might worsen atherosclerotic vascular damage and in turn, hypertension through small VLDL and LDL deposition in vessels intima [29].

Chronic kidney disease (CKD) also plays a role in the pathogenesis of obesity and hypertension as increased visceral adiposity is associated with impaired kidney function through physical compression of the kidneys by the fat around them, activation of the renin-angiotensin pathway, as well as increased sympathetic nervous system activity. Constriction of the efferent arteriole with an increase in the intraglomerular pressure leads to nephron loss and increased renal tubular sodium reabsorption, which in turn impairs pressure natriuresis plays a crucial role in the development of increased blood pressure in obese individuals [30].

Hyperinsulinemia and insulin resistance often present in obese individuals play as well a significant role in the genesis of hypertension [31]. Several studies in experimental animals and humans have provided evidence that hyperinsulinemia may elicit increases in sympathetic nervous system activity, activate the renin-angiotensin system, and increase renal sodium retention that, if sustained, could increase blood pressure [32]. Insulin acts on almost all of the nephron segments by activating transporters such as the Sodium-proton exchanger type 3 (NHE3), the main regulator of sodium reabsorption in the proximal tubule, and the epithelial sodium channel (ENaC) in the distal nephron and the connecting tubule, another important contributor to sodium reabsorption [33]. In addition, w-no-lysine (WNK) kinases, responsible for familial hypertension, through the stimulation of sodium reabsorption in the distal nephron, have also been found to be regulated by insulin [34].

Sleep breathing disorders and sleep apnea are also extremely common in obese patients, with a prevalence of 40 to 90% [35]. Sleep apnea is a known cause of hypertension through neurohormonal dysregulation, endothelial dysfunction, inflammation, and increased levels of endothelin by repeated episodes of hypoxia [36].

Finally, the leptin-melanocortin pathway has been extensively studied in animal models as a driving factor of hypertension in obesity. Leptin is a protein that signals to the brain the amount of stored fat, and it is speculated that leptin exerts negative feedback on brain centers involved with energy intake. The hypothesis behind obesity-induced hypertension involves the downstream effects of hypothalamic leptin signaling and the activation of specific melanocortin receptors located on sympathetic neurons in the spinal cord. The physiological consequences of this sympathetic activation are activation of the renin-angiotensin system, sodium retention, circulatory expansion, and elevated blood pressure [37].

## 3. Genetics, Epigenetics and Mitochondrial Factors Related to Obesity and Hypertension

### 3.1. Genetics

A major factor in the interplay between obesity and hypertension is genetic susceptibility. A recent population-based study on 30,617 twin individuals showed that being overweight or obese was associated with a 94% increased risk of hypertension (OR = 1.94, 95% CI: 1.64~2.30). After controlling for all other variables, this association was mainly explained by genetics [38]. However, there is still no consensus on which specific genes have a direct role in both obesity and hypertension. Currently, several molecular mechanisms (both genetic and epigenetic) are under investigation to elucidate the role of genetics in the pathophysiology of obesity-related hypertension. This role has been primarily investigated by identifying single nucleotide polymorphisms (SNPs) in loci associated with both diseases. Genome-wide association studies (GWAS) have identified more than 50 SNPs associated with hypertension and over 250 genes/loci involved in the pathophysiology of obesity [39].

Notably, Melka et al. showed that three previously identified loci of obesity (rs16933812 on Paired box protein Pax-5 [*PAX5*], rs7638110 on Mitochondrial Ribosomal Protein S22 [*MRPS22*], and rs9930333 on Fat Mass and Obesity-associated [*FTO*]) were also associated with elevated systolic blood pressure in an adolescent Canadian population [40]. A meta-analysis of 98,720 patients (57,464 hypertensive cases and 41,256 controls) also identified an association between *FTO* gene variants and hypertension in obese individuals and more so in Asians [41]. However, more recent studies have not been able to verify those results. A replication study by Goulet et al. was done on a similar sample population and using similar methods as the study by Melka et al. Their results showed that rs9930333 (*FTO*) was associated with increased body mass index, but not with increased systolic blood pressure [38]. A cross-sectional study also found no significant difference in genotypic frequencies of *FTO* rs9939609 SNP between hypertensive patients and healthy controls [42]. The discrepancies of those results in the association of *FTO* with hypertension suggest that there may be phenotypic and genetic differences across populations. Further studies with higher power should be conducted to eliminate the possibility of selection bias and false positivity that may result from selecting specific nucleotide variants. 

Family studies have as well correlated obesity with cardiovascular risk factors such as hypertension. This familial correlation emphasizes the importance of pleiotropic genes that control more than one trait at a time [39]. For instance, the Oman family study (n = 1231) showed that all obesity indices (BMI, percentage body fat, waist circumference, and waist-to-height ratio) had consistently positive phenotypic correlations with ambulatory and office beat-to-beat SBP and DBP. Genetic correlations of obesity indices with hypertension were also higher than environmental correlations [43]. However, there is still no full agreement on this familial association of adiposity with blood pressure. For instance, the Quebec Family Study showed that only a moderate association exists between adiposity and blood pressure, with both genetics and the environment playing a role in this correlation [44]. Moreover, there was no evidence of pleiotropy between measures of resting blood pressure and body composition in the HERITAGE Family Study [45].

### 3.2. Epigenetics

The rising rate of obesity in recent years cannot be explained by genetics alone. Epigenetic factors that alter gene expression without interfering with DNA structure play an important role in this trend. Three epigenetic modifications have been studied in relation to obesity and hypertension: (a) DNA methylation, (b) histone modification, and (c) non-coding RNA. DNA methylation is the most studied [46] and consists of the covalent binding of a methyl group to a cytosine residue in the DNA, at sites where cytosines are followed by guanines (CpG sites). This process is mediated by methyltransferases and its dysregulation (either hypermethylation or hypomethylation) can alter gene transcription and lead to several diseases [47].

The association between methylation and obesity has been inconsistent in studies, especially with global methylation. Some studies associated excess adiposity with global DNA hypermethylation, while others associated it with global hypomethylation [48]. However, results from specific gene methylation (candidate genes) are more congruous. The genes that are regulated by methylation and studied in the context of the pathogenesis of obesity include leptin gene (*LEP*), adiponectin gene (*ADIPOQ*), peroxisome proliferator-activated receptor γ coactivator 1 alpha gene (*PGC1A*), insulin-like growth factor-2 (*IGF-2*), and hypoxia-inducible factor 3a (*HIF3A*) [47]. In hypertension, epigenetic mechanisms also play a role through the methylation of different genes. A systematic review of the role of DNA methylation in modifying blood pressure showed that lower methylation levels of *SULF1* (sulfate endosulfatase)*, EHMT2* (Euchromatic Histone Lysine Methyltransferase 2), and *SKOR2* (SKI Family Transcriptional Corepressor 2) were associated with hypertension. On the other hand, lower methylation levels of *PHGDH* (phosphoglycerate dehydrogenase), *SLC7A11* (Solute Carrier Family 7 Member 11), and *TSPAN2* (Tetraspanin 2) were correlated with higher systolic and diastolic blood pressure [49].

Another important epigenetic mechanism is histone modification. Histones are proteins wrapped around DNA that contribute to making compact chromatin. Histone modification (acetylation, deacetylation, methylation, phosphorylation, or ubiquitination) is one of the epigenetic mechanisms that emerged as a factor in obesity and energy metabolism [50]. Recent studies have also investigated the role of histone modification in obesity-related hypertension. For instance, Jung et al. found that a high-fat diet inhibits the MsrA (Methionine Sulfoxide Reductase A)/Hydrogen Sulfide (H2S) Axis, which causes oxidative stress, inflammation, hyper-contractility, and hypertension. At the same time, MsrA/H2S is epigenetically regulated through histone deacetylation. In the same study, Jung et al. showed that inhibiting histone deacetylases (HDACs) improved obesity-induced hypertension through the restoration of the MsrA/H2S axis. This finding could place HDACs as potential therapeutic targets for hypertension in obesity [51]. 

MicroRNAs (miRNAs) are small, single-stranded, non-coding RNA molecules that regulate post-transcriptional gene expression. They function by silencing messenger RNA (mRNA) by binding to its 3′ untranslated region [52]. MicroRNAs play an important role in both physiologic and pathologic processes. Dysregulation of mRNAs has been associated with obesity and obesity-related inflammation. In fact, the metabolic disruption in obesity is rooted in chronic low-grade inflammation through the activation of TNFα, IL-6, and CRP [53]. MicroRNAs modulate this process by controlling the post-transcriptional expression of those cytokines [54]. Several studies have linked shifts in mRNA expression to phenotypes of obesity and metabolic syndrome [55]. In fact, 221 out of 1736 loci of obesity in an online database (integratomics) corresponded to micro-RNAs [56]. Several studies have investigated how dysregulation in those miRNAs affects obesity phenotypes. For example, Hijmans et al. found that circulating miR-34a levels were significantly higher in the obese as compared with normal weight and overweight groups, while miR-126, miR-146a, and miR-150 levels were significantly lower in both the obese and overweight groups [57]. Pan et al. also highlighted the role of adipocyte-secreted microRNA-34a (miR-34a) in inhibiting polarization of the anti-inflammatory M2 phenotype of macrophages in adipose tissues of obese subjects, which contributes to the obesity-induced systemic inflammation and metabolic dysregulation [58].

Other miRNA dysregulations that were associated with human obesity and obesity-related inflammation include miR-33, miR-22, miR-29a, miR-221, miR-30, miR-26b, miR-199a, and miR-148a [59,60,61,62,63]. However, few studies in the literature link miRNAs to obesity-induced hypertension. One of those studies was conducted on a cohort of several obese children, and it showed that higher levels of miR-192 were detected in obese patients with essential hypertension compared to those with normal blood pressure [64]. Another case-control study concluded that there were significant variations in levels of miRNA-122 and miRNA-33 in the group of central obesity and hypertension compared to the control group [65]. In addition, Eikelis et al. found that the expression of miR-132 in subcutaneous fat is associated with higher blood pressure in obese patients [66]. Further studies should elucidate the exact role of specific microRNAs in the pathophysiology of obesity-induced hypertension because those microRNAs can serve as diagnostic and prognostic markers for metabolic disorders, including obesity and hypertension.

### 3.3. Mitochondria

Aside from genetics and epigenetics, mitochondria also play a role in the multifactorial etiology of obesity-induced hypertension. Indeed, obesity-associated oxidative and inflammatory stress has been linked to a variety of cardiovascular diseases through the development of metabolic syndrome. This pathophysiologic process involves the abnormal production of reactive oxygen species (ROS) that leads to mitochondrial dysfunction [67]. In normal cells, mitochondria are highly dynamic cytoplasmic organelles that play an essential role in cellular metabolism through four dynamic processes: (a) fusion of two mitochondria into one, (b) fission or division of mitochondria into two or more; (c) biogenesis, which is required for cell growth and adaptation, and (d) mitophagy, a specialized form of autophagy similar to cell apoptosis [68].

Mitochondrial dysfunction through disruption of those dynamic processes contributes to oxidative stress and is associated with the development of both obesity and hypertension. The equilibrium between mitochondrial fusion and fission is modulated through nutrient availability and metabolic demands [69]. Nutrient depletion triggers the “Stress-Induced Mitochondrial Hyperfusion” or SIMH response, which leads to mitochondrial fusion. SIMH is an adaptation response to stress because it leads to increased ATP production and NF-κB activation, which in turn protects against autophagy and apoptosis. SIMH is essentially mediated through fusion machinery proteins (*MFN1* [Mitofusin-1] and *OPA1* [Mitochondrial Dynamin Like GTPase]), as well as the scaffold protein stomatin-like protein 2 (SLP2) [70]. Nutrient depletion also stops mitochondrial fission through the PKA-mediated phosphorylation of DRP1 [71].

On the other hand, obesity and adiposity blunt mitochondrial fusion and triggers mitochondrial fragmentation [69]. This has been demonstrated in both human and animal models. For instance, Dietrich et al. showed that in overweight rodents, the high-fat diet induces mitochondrial fission and diminishes the expression of MFN2 in the hypothalamus [72]. In humans, *MFN2* (Mitofusion-2) expression is decreased in the skeletal muscle of obese and diabetic patients, and this effect is reversed after weight loss [73].

In hypertension, activation of the sympathetic nervous system is a key element in cardiac remodeling, especially in obese patients. This effect is mediated through the release of catecholamines (i.e., norepinephrine) which induce hypertrophy of the cardiac myocytes. These myocytes are abundant in mitochondria since they rely on them for a constant supply of ATP needed for the cyclic contractions of the heart. In this regard, hypertension-induced cardiac hypertrophy causes dysfunction of ATP production and the electron transport chain in the mitochondria. In addition, mitochondrial dysfunction in the cardiac myocytes of hypertensive patients is linked to disrupted dynamics of fusion and fission. For instance, the expression of *OPA1*, *MFN1*, and *MF2* was decreased in hypertensive rats, all together favoring mitochondrial fission [69].

Moreover, both mitochondrial tRNA 15910 C > T mutation and Gly482Ser polymorphisms in PGC-1α (Pparg coactivator 1 alpha) were associated with essential hypertension [74,75]. Considering that both hypertension and obesity have underlying pathophysiologic processes related to mitochondrial dysfunction, an inevitable question arises: Could the disruption of the mitochondrial dynamics in adiposity also cause the development of hypertension in obese patients? A recent study published by Li et al. found that the knockout of *TRPV1* (nonselective cation channel) and *UCP*1 (uncoupling protein) in mice at the same time induced severe obesity and obesity-associated hypertension. Hypertension was the result of disruption of mitochondrial calcium uptake and subsequent ROS production in brown adipose tissue [76].

Another relevant hypothesis that helps answer the question was generated by investigating the role of leptin in mitochondrial dysfunction [69]. Kamareddine et al. argue that leptin is an obesity-associated pro-inflammatory adipocytokine whose elevated levels of obesity are associated with acute cardiovascular events and obesity-related hypertension [77]. Indeed, leptin secretion by adipose cells leads to mitochondrial dysfunction within the white adipose tissue. This association is also bidirectional: mitochondrial dysfunction within the white adipose tissue leads to the overproduction of leptin. Consequently, this generates a self-reinforcing cycle of leptin secretion and mitochondrial dysfunction. Leptin can also activate the sympathetic nervous system and the release of catecholamines previously discussed. All those pathologic processes combined can eventually contribute through mitochondrial dysfunction to metabolic and cardiovascular consequences, including hypertension.

## 4. Characteristics of Hypertension in Obese Patients

The diagnosis and monitoring of hypertension in obesity is often complicated by difficulties in accurately measuring blood pressure in these patients, as a standard cuff is inappropriate in most [78]. The American Heart Association recommends using a cuff size of 20 × 42 cm bladder and 45–52 cm arm circumference to obtain precise values [79]. The BMI shows the highest strength in association with high BP among all anthropometric indicators of obesity, such as body adiposity index, waist circumference, and weight-height ratio. For males, the probability of the elderly presenting high BP increased by 2% for every unit increase in the BMI [80].

Although the investigation of secondary causes for early hypertension is generally recommended, it may not always be necessary for obese patients as obesity is associated with higher blood pressure numbers in pediatric and adolescent populations [81,82]. A rational approach to investigation across strata of age, body mass index (BMI) sex, and race, based on BP distributions in the US National Health and Nutrition Examination Surveys were conducted in 2005 to 2016 and used to define the threshold for investigation of a secondary cause in a US population at thresholds ranging from ≥130/≥80 and ≥140/≥90 for normal BMI women and men, respectively, at age 20 to 30 years, to ≥160/≥100 and ≥170/≥105 for women and men with BMI ≥ 40 at age 30 to 40 years [83].

Patients with obesity and hypertension present characteristic hemodynamic changes including high cardiac output, high plasma volume, and paradoxically normal total peripheral resistance that makes them more likely to develop left ventricular hypertrophy and kidney damage than observed in lean hypertensive patients [84]. However, Sokolow-Lyon Electrocardiography voltage criteria, for example, are significantly less sensitive in detecting left ventricular hypertrophy in obese individuals [79]. Regarding the management of obesity-related hypertension, the pharmacokinetics, and pharmacodynamics of many medications are also impacted by excess adiposity with abnormal drug handling, expanded volume of distribution, and altered hepatic and renal clearance [85], rendering the dosing of medications often challenging.

## 5. Treatment of Obesity to Control Hypertension

### 5.1. Current Treatments

Weight loss is a crucial step and a lifelong process for tackling obesity-associated diseases, and its association with BP control is now well documented [86]. Lifestyle modifications that include dietary regimens, an increase in physical activity, and cognitive-behavioral therapy represent the first step of the process. As the results are often suboptimal, effective complementary steps include drug therapy, bariatric surgery [87], and experimental therapies.

#### 5.1.1. Diet

Multiple diet options have been studied with various success rates in achieving and maintaining weight loss. The common goal is to maintain a status of caloric deficiency and create an incremental change in the body’s composition [88]. Notably both the Dietary Approaches to Stop Hypertension (DASH) diet, which consists of plant-based food and dairy products low in fat [89], and the Mediterranean diet have been successful not only in reducing both systolic and diastolic blood pressure but also in decreasing obesity-associated inflammation and complications [90]. The green Mediterranean diet, a version of the Mediterranean diet amplified with green plant-based proteins/polyphenols such as green tea and walnuts, and restricted in red/processed meat, is particularly indicated to reduce intrahepatic fat content [91]. Another promising regimen is the Very Low-Calorie Ketogenic Diet (VLCKD). This program restricts the caloric content to only 500–800 calories per day, with significantly lower carbohydrates (<50 g/day) in meals. VLCKD showed significant but equal results not only in lowering body weight and waist circumference but also in enhancing the participants’ lipid and glucose values [92]. The greater the weight loss, the more significant the improvements in cardiovascular health and blood pressure parameters [93].

#### 5.1.2. Physical Activity

An increase in physical activity is an important adjunctive to the right diet in the process of achieving the target body weight and normalizing blood pressure. It enhances the patients’ mood and strengthens their commitment to the dietary regimen [94]. Aerobic exercises have shown the best outcome in decreasing total body weight and fat percentage, whereas resistance training is the method of choice to grow lean body mass [95]. Patients that combine both regimens have lower pervasiveness of obesity [96]. The current recommendation is 30 min or more of aerobic exercises on most days, along with 30–45 min of strength training at least 3 times a week [97]. Even in the absence of weight reduction, regular workouts have proven to significantly decrease visceral, abdominal, total body, and skeletal muscle fat [98]. The greater the drop in weight circumference, the greater the improvement in blood pressure and other metabolic syndrome components [93].

#### 5.1.3. Drug Therapy

Pharmacotherapy is a supplemental tool to lifestyle modifications and is being increasingly prescribed. It is advised according to the European Association for the Study of Obesity in patients whose BMI is ≥30 kg/m^2^ or BMI ≥27 kg/m^2^ with obesity-related comorbidity such as hypertension [99]. Physicians should use constant vigilance in assessing the patient’s response to pharmacotherapy. Medications should be stopped or changed if the side effects are intolerable, or the weight loss is less than 3% in diabetics or 5% in nondiabetics [99].

Pharmacological treatment has been increasingly versatile, with combinations tailored to the patients’ profiles. The most promising class is the class of incretins-mimetics. Incretins such as glucose-dependent insulinotropic polypeptide (GIP) and glucagon-like peptide 1 (GLP-1) are factors released by the gut in response to ingested nutrients and play an important role in metabolism as they regulate appetite and body weight [100]. GLP-1 receptor stimulation can decrease appetite and food intake [101,102], as well as gastrointestinal motility and gastric emptying [103,104]. These effects have led to the development of GLP-1 receptor (GLP-1R) agonists not only to treat diabetes but also to decrease adipose tissue mass and to be used for weight loss [102]. Among them, subcutaneous Semaglutide seems to be the most efficacious in reducing body weight, followed by oral Semaglutide, exenatide twice daily, and liraglutide [105]. For instance, in phase III clinical trials conducted in individuals with obesity or overweight without diabetes, Semaglutide at the dose of 2.4 mg led after 68 weeks of treatment to a decrease in body weight by −14.9% relative to −2.4% in placebo-treated controls [106]. In addition, GLP-1R agonists were found to be effective at decreasing blood pressure, with Semaglutide demonstrating again the most effectiveness [105,107,108,109]. This substantial effect on blood pressure might be attributed to the GLP-1 enhancement of natriuresis [101,110]. On the other hand, GIP receptor (GIP-R) agonists were found to be either weight neutral or to induce modest and dose-dependent weight loss in mice, and when combined with long-acting selective individual agonists for GLP-1R, to enhance body weight lowering [111,112,113,114]. Subsequently, the use of dual GLP1-R and GIP-R agonism has been studied and shown to reduce body weight by more than 20% [115,116]. A new drug with that dual agonism, Tirzepatide, was approved by the United States Food and Drug Administration (FDA) in May 2022 and was shown in a phase 3 open-label study to be superior to Semaglutide for body weight reduction [117,118].

Another anti-diabetic agent pramlintide, an amylin analog that is approved by the FDA for use in patients with diabetes who are using mealtime insulin alone, or in combination with an oral agent, such as metformin or a sulfonylurea [119], is also showing promise for weight loss and its usefulness is not limited to patients with impaired glucose metabolism [120]. Amylin is a peptide that is co-secreted with insulin and reduces food intake through central control of satiety pathways [121].

Other commonly used therapies include the combination of phentermine and Topiramate [94] and the naltrexone bupropion combination [119]. Phentermine belongs to the sympathomimetic amines family and targets norepinephrine (NE) and dopamine (DA) neurons in the brain [122], whereas topiramate is a known anticonvulsant which has been noticed initially in clinical trials to induce weight loss in trials for seizure disorders and was thereafter tried for this indication [123]. Clinical trials such as the CONQUER, the SEQUEL, and the EQUIP studies have successfully tested this combination and showed clear benefits to blood pressure and weight in a dose-dependent fashion [124,125,126] without compromising the cardiovascular health of patients in the long term [127]. Regarding the naltrexone-bupropion combination, bupropion is a known atypical antidepressant with positive effects on smoking cessation [128], while naltrexone is an opioid antagonist used for alcoholism and opioid use disorder treatment [129]. The concurrent use of both medications affects the brain, mainly the hypothalamic melanocortin system, to suppress appetite in a not completely understood mechanism yet [130]. Nonetheless, as this combination could raise blood pressure, its use is discouraged in the setting of concomitant obesity and hypertension [131]. 

Table 1 summarizes the characteristics of the current agents approved for the treatment of hypertension. In the case of failure of diet, exercise, and obesity medications in achieving target blood pressure or the presence of multiple risk factors for complications in obese patients, typical antihypertensive drugs should be considered. First-line treatment for overweight hypertensive patients should include one of the following: a long-acting calcium channel blocker, an angiotensin-converting enzyme inhibitor, or an angiotensin receptor blocker because of its neutral effect on weight [29]. Telmisartan might be superior in the treatment of obesity-related hypertension due to its role in countering adipose tissue’s negative effects in an experimental model [132].

#### 5.1.4. Bariatric Surgery

Bariatric Surgery is traditionally indicated for patients who have failed the initial interventions and have a BMI of 35 kg/m^2^ or more and an obesity-related comorbidity or a BMI more than 40 kg/m^2^ with or without the presence of comorbidities even if the indications have been recently evolving as people with BMI 30 to 35 with type 2 diabetes mellitus could also be considered [143].

The 2 most common procedures used currently, the sleeve gastrectomy and gastric bypass, have similar effects on weight loss and a similar safety profile through at least 5-year follow-ups [144]. Massive weight loss secondary to bariatric surgery can trigger profound sympathoinhibitory effects and is associated with a stable and significant reduction in plasma leptin levels with subsequent blood pressure reduction [145]. In a series of 45 patients, postoperative weight loss after gastric bypass surgery was associated with the resolution or improvement of diastolic hypertension in approximately 70% of cases [146]. Regarding the best technique, gastric bypass surgery seems to be superior to gastroplasty and gastric banding at achieving and sustaining lower systolic and diastolic blood pressure and has also shown a better effect on natriuresis [147]. 

### 5.2. Experimental Pathways 

#### 5.2.1. Molecular Regulators of Adipogenesis

3-Amino-1,2,4-triazole (ATZ) is a heterocyclic organic compound that inhibits *α*-oxidation, fatty acid synthesis, and lipogenesis in isolated hepatocytes [148]. ATZ also inhibits aminolevulinic acid dehydratase, a key enzyme in heme synthesis. As heme activates the transcription repressor RevErb*α*, which is essential for adipocyte differentiation, ATZ could also potentially inhibit adipogenesis via that route [149]. In a group of mice fed with a high-fat diet (HFD), the administration of ATZ over 12 weeks led to the prevention of an increase in blood pressure and lower body weight, triglycerides levels, and leptin in plasma. ATZ treatment also impeded an HFD-induced increase in adipocyte diameter and induced marked atrophy and the accumulation of macrophages in this tissue [150], therefore, showing promise as a potential agent to be used in the future in the treatment of hypertension in obesity in particular and metabolic syndrome in general.

IL-20, a pro-inflammatory cytokine, may play a significant role in obesity, with levels elevated in obese states and decreasing with weight loss [151]. IL-20 may also play a significant role in hypertension as it belongs to the structurally and functionally similar IL-10 family [152]. IL-10 gene knockout mice fed with a high salt diet had lower blood pressure, lower response to angiotensin II, and higher nitric oxide compared to wild-type mice [153].

The therapeutic potential of a targeted monoclonal antibody 7E to IL-20 for decreasing diet-induced obesity was tested in murine models [150] (p. 20). The experiment found that IL-20 levels were higher in obese mice who were leptin-deficient, and leptin-resistant, and in those who were subjected to an HFD (60% kcal derived from fat) compared with a low-fat diet (LFD)-fed (10% kcal derived from fat) controls. IL-20 expression was also found to be higher in adipocytes and adipose tissue macrophages of visceral WAT of HFD-mice compared to LFD-mice. On the other hand, treatment of mice with 7E antibody resulted in reduced body weight, improved glucose metabolism and insulin resistance, lower fat pad weight and serum triglyceride levels, lower transaminase levels, and hepatic steatosis as compared with mice treated with nonspecific monoclonal IgG antibodies (mIgG). Furthermore, adipose tissue M1 macrophages of 7E-treated mice were fewer in number than those of mIgG-treated mice. Thus, targeting IL-20 could provide direct and indirect treatment for hypertension through weight loss.

#### 5.2.2. Incretins Pathway

GLP-1 and GIP agonism has been successfully used to lower weight as detailed above. Conversely, it has also been noticed that disruption of the GLP-1 signaling in mice was protective against obesity [154]. In addition, the deletion of the GIP-R in mice was also found to have the same effect [155,156,157]. Subsequently, the effect on weight gain of blocking GIP-R and the GLP-1R with targeted antibodies, in both healthy and obese mice was studied [158]. The experiment showed reduced food intake and weight gain in mice when either anti-GIPR or GLP-1R antibodies were used. Furthermore, the effect was additive when both receptors were antagonized.

This paradoxical effect has created some perplexity in the scientific community, however, two hypotheses have emerged to reconcile agonism with antagonism [159]. The first one postulates that eliminating a signaling axis involved in the metabolic response to food intake seems to enhance other systems in the energy balance network. The second one supposes that chronic agonism of the GIPR produces desensitization of the GIP system and ultimately the same result as a GIP-R antagonist.

The effect of GLP-1 agonists on blood pressure in humans is not completely understood yet. The majority of the studies done on mice showed a positive effect on BP through ANP secretion and natriuresis [160]. While research has shown a positive correlation between GLP-1 levels and both systolic and diastolic pressure in healthy adults, acute administration of GLP-1a to healthy adults has shown no changes in blood pressure [160,161]. However, GLP might indirectly reduce BP through weight loss.

#### 5.2.3. Leptin and Leptin Sensitizers

Leptin, a hormone secreted by the adipose tissue, plays a major role in body weight homeostasis and helps reduce food intake and increase energy expenditure [162]. As congenital leptin deficiency was found to result in severe weight gain and obesity in humans [163], leptin was considered an important regulator of energy balance, and its potential role as a therapy to reduce obesity was explored. However, administering additional leptin in the context of obesity has been largely ineffective, as obese individuals have higher circulating levels of leptin along with leptin resistance [164,165].

Conversely, a decremental decrease in circulating leptin levels in adult obese mice was found to initiate an unexpected and significant improvement in several parameters of energy balance and glucose homeostasis [166]. In the same experiment, the treatment of obese mice with leptin-neutralizing antibodies also led to reduced food intake. These findings led another group to use successfully recombinant yeast for in vivo protein interference, a protein knock-down strategy for partial leptin reduction and weight loss. 

Chronically high Leptin levels in obese patients lead to sodium retention secondary to increase sympathetic tone and low nitric oxide [167]. As Leptin is a key player in the pathogenesis of hypertension in obesity, strategies aimed at partially reducing circulating leptin may represent a promising approach for the treatment of obesity and hypertension [168]. Some angiotensin receptor blockers (ARB) and angiotensin-converting enzyme (ACE) inhibitors like candesartan and enalapril have already achieved this purpose in practice by lowering leptin, BP, and weight [167].

Another strategy to enhance leptin activity and make it an effective agent for weight loss would be to combine it with another molecule with the achievement of an enhanced effect and subsequent weight loss. Examples include pramlintide [169,170], exendin 4 and fibroblast growth factor 21 (FGF21) [171]. A promising combination for the future is Pramlintide-metreleptin [172].

#### 5.2.4. Amylins

Besides pramlintide, other amylin analogs with improved pharmacokinetics are being considered as potential therapeutic agents, and the amylin pathway is another very active area of experimental investigation. For instance, Cagrilintide, a long-acting amylin analog has made it successfully to a phase II trial [173], and concomitant treatment with Cagrilintide and Semaglutide was well tolerated in a phase 1b trial, opening pathways to potential new combinations to be used. Also, Amylin acts on several receptors, and the subtypes are complexes of the calcitonin receptor with receptor activity-modifying proteins [174]. Therefore, amylin agonists can have enhanced efficacy when combined with calcitonin receptor agonists, and a new potential class is the Dual acting Amylin and Calcitonin receptor agonists (DACRA) with the ability to act as insulin sensitizers [175].

Animal studies have demonstrated that DACRAs possess enhanced metabolic efficacy as compared to amylin regarding weight loss, and they also improve post-prandial glucose control [176,177,178]. In the recent STEP trials, Semaglutide showed a significant reduction in blood pressure and weight in obese patients [179].

Most recently, the acylated form of KBP-066A, a long-acting dual amylin, and calcitonin receptor agonist, has shown increased potency and efficacy to the non-acylated form in rats showing promise for this molecule as both anti-obesity, anti-hypertensive anti-diabetic agent [180].

#### 5.2.5. Ghrelin

Ghrelin is a peptide hormone secreted from the gastric fundus that acts on receptors in the Hypothalamus to stimulate food intake in a dose-dependent fashion [181]. Ghrelin is found in 2 forms: Acyl-Ghrelin and Desacyl-Ghrelin, however, only the acylated form (which is acylated by the Ghrelin O-Acyltransferase enzyme) binds the Ghrelin receptor, which is the growth hormone secretagogue receptor (GHSR) to exert its effect [182,183]. Conversely, the liver-expressed antimicrobial peptide 2 (LEAP2) acts as an antagonist of ghrelin by inhibiting GHSR activation [184]. Both plasma levels of ghrelin and LEAP2 are highly regulated by body weight and feeding status in opposite directions [185]. Agents that act to lower plasma ghrelin, raise plasma LEAP2, block GHSR activity, and/or raise desacyl-Ghrelin signaling could therefore be efficient to treat obesity. Furthermore, ghrelin has a direct positive effect on blood pressure by decreasing sympathetic activity and stimulating the production of nitric oxide. It has been documented to lower blood pressure in both healthy and hypertensive human and animal subjects [186].

#### 5.2.6. Mitochondrial Therapies

Normally, nutrient oxidation and ATP synthesis are linked together via a mechanism in which the oxidation of nutrients powers proton pumps in the mitochondrial membrane to eject protons out of the mitochondria matrix, followed by re-entry to drive ATP synthesis. When proton transport into the mitochondrial matrix stops being linked to ATP synthesis, this process is known as uncoupling [187]. Humans possess the uncoupling protein UCP1, localized inside the brown adipose tissue, and that works to uncouple O2 consumption and ATP synthesis [188]. Mitochondrial uncouplers have been investigated to decrease caloric efficiency with more nutrients being oxidized to produce a given amount of ATP as a possible mechanism of weight reduction [189,190] (p. 10).

The first promising agent BAM15((2-fluorophenyl)(6-[(2-fluorophenyl)amino](1,2,5-oxadiazolo [3,4-*e*]pyrazin-5-yl))amine) a mitochondria-specific protonophore uncoupler, that is orally bioavailable prevented and reversed diet-induced obesity and insulin resistance in mice without altering food intake or decreasing lean body mass [191].

A more potent derivative of BAM15 named SHC517 (N5–(2-fluorophenyl)-N6 -(3-fluorophenyl)-[1,2,5]oxadiazolo-[3,4-b]pyrazine-5,6-diamine) was also tested in a mouse model and administered as an admixture in food [187]. It increased lipid oxidation without affecting body temperature, prevented diet-induced obesity, and reversed established obesity. In addition, it improved glucose tolerance and fasting glucose levels. Importantly, the drug was not found to affect food intake or lean body mass.

Significant research needs to be done to have a better understanding of the roles that mitochondria and mitochondrial uncoupling play in the pathogenesis of obesity and hypertension. Some evidence suggests that mitochondrial UCP2 is upregulated in oxidative stress. This might have a protective role against the development of hypertension [192]. Moreover, the use of mitochondrial-targeted antioxidants such as MitoQ and MitoTEMPO not only provides primary prevention against hypertension but also alleviated cardiac hypertrophy and heart failure secondary to high afterload [190,193,194]. In addition, the combined use of mitochondrial antioxidants and regular hypertensive pharmacotherapy shows a synergistic effect on BP [194].

#### 5.2.7. GDF15

Growth differentiation factor 15 (GDF15) is a stress-regulated hormone that is normally found at low levels but is increased in inflammation and chronic diseases such as cardiovascular disease and cancer [195]. GDF15 acts on the GFRAL receptor in the hindbrain and leads to a dose-dependent decrease of food intake in rodents [196,197,198]. One of the effects of GDF-15 includes triggering visceral malaise, a phenomenon like food aversion seen in sickness. Both acute and chronic GDF15 exposure are able to trigger visceral malaise, demonstrable by increased ingestion of non-nutritive food [199] (p. 15). This response may be used to decrease caloric intake in obesity and drive weight loss. Another interesting finding is related to exercise as intense physical activity can increase GDF-15 levels in both mice and humans [200,201]. This finding is possibly related to the fact exercise can be seen as a stressor on the body [202]. GDF15 is consistently elevated in arterial hypertension [203].

Cisplatin increases GDF-15 levels by around 3-fold, and one of the well-known side-effects of Cisplatin use is emesis and anorexia. Therefore, this side effect of Cisplatin could be accounted for by the rise in serum GDF-15 levels, giving us more insight into possible future uses of this hormone [204,205].

#### 5.2.8. Peptide Tyrosine Tyrosine

Peptide Tyrosine Tyrosine (PYY) is a peptide hormone co-secreted by intestinal L cells as PYY1-36 along GLP-1 in response to nutrient intake and cleaved into its active form PYY3-36 by DPP-IV [206]. PYY3-36 acts on NPY receptor type 2, expressed centrally including limbic and cortical areas and peripherally [207]. PYY3-26 plays an important role in energy homeostasis as its administration has been shown to lead to decreased food intake in both humans and rodents and to reduce body weight in rodents [208,209]. It does so through silencing Npy neurons and, hence, indirectly activating Pomc neurons [209], but also through activation of the mesolimbic dopaminergic system as well as of GABAergic and glutamatergic neurons in cortical and subcortical regions and the brainstem [207]. PYY3-26 agonists are subsequently being developed and tested as anti-obesity agents. In a rat model, Roux-en-Y gastric bypass (RYGB) and liraglutide+PYY_3-36_ induced a similar body weight loss, however only RYGB induced significant metabolic improvements [210]. PYY does not appear to have a direct role in the treatment of hypertension but might promise a reduction in blood pressure through its anti-obesity effects.

## 6. Conclusions

Hypertension is closely linked to the prevalence, pathophysiology, and morbidity of obesity. The pathogenesis of hypertension related to obesity is multifactorial and complex. Weight loss stabilizes neurohormonal activity and causes clinically significant reductions in blood pressure. Bariatric surgery remains so far, the most reliable method to achieve sustained weight reduction. However, new drugs such as Semaglutide and Tirzepatide are also emerging as potent, safe, and effective alternatives for weight reduction. In addition, many new investigational drugs that could potentially also alleviate the morbidity and mortality associated with obesity-induced hypertension are being developed and their role should be further defined in the future.

## Figures and Tables

**Table 1 ijms-23-12305-t001:** FDA Approved medications for weight management.

Medication	Mechanism	FDA Approval Date	Contraindications	Side Effects	Further Information
Tirzepatide (MOUNJARO) [118]	Glucagon-like peptide-1 agonist	2022	Personal or family history of medullary thyroid carcinoma or in patients with Multiple Endocrine Neoplasia syndrome type 2	Nausea, diarrhea, decreased appetite, vomiting, constipation, dyspepsia, and abdominal pain	Indicated as an adjunct to diet and exercise to improve glycemic control in adults with type 2 diabetes mellitus.
			Known serious hypersensitivity to Tirzepatide or any of the excipients in MOUNJARO		
Semaglutide (Wegovy) [133,134]	Glucagon-like peptide-1 agonist	2021	In combination with other weight-loss agents	Nausea, diarrhea, vomiting, constipation, abdominal pain, dyspepsia, flatulence, abdominal distension,	The only FDA-approved drug for chronic weight management in adults with general obesity or overweight since 2014
			Personal or family history of medullary thyroid carcinoma	hypoglycemia in type 2 diabetics, headache, fatigue	Indicated for patients with BMI of 27 kg/m^2^ or more with at least one weight-related ailment or a BMI of 30 kg/m^2^ or greater.
			Patients with Multiple Endocrine Neoplasia syndrome type 2 (MEN 2)		Not studied in patients with a history of pancreatitis
Liraglutide (Saxenda) [135,136]	Glucagon-like peptide-1 (GLP-1) receptor agonist	2014 for adults	Personal or family history of medullary thyroid carcinoma	Nausea, vomiting, hypoglycemia, diarrhea, constipation, abdominal pain, dyspepsia, headache,	Indicated for pediatrics with BMI cut-offs for age and sex that correspond to a BMI 30 kg/m^2^ or higher for adults, and who weigh more than 60 kg
		2020 for pediatrics 12 or older	Patients with MEN 2	fatigue, dizziness, hypoglycemia, and increased lipase	Indicated for adults with a BMI of 27 kg/m^2^ or more with at least one weight-related ailment or a BMI of 30 kg/m^2^ or greater.
					Use for pediatrics with type 2 diabetes not established
Phentermine and topiramate extended-release capsules (Qsymia) [137,138]	Phentermine, a sympathomimetic amine anorectic, and topiramate	2012 for adults	Pregnancy. Risk for fetal malformations (cleft lip and cleft palate)	Adults: paresthesia,	Indicated for pediatrics with BMI cut-offs for age and sex that correspond to a BMI 30 kg/m^2^ or higher for adults, and who weigh more than 60 kg
		2022 for pediatrics 12 or older	Glaucoma	dizziness, dysgeusia, insomnia, constipation, and dry mouth	Indicated for adults with a BMI of 27 kg/m^2^ or more with at least one weight-related ailment or a BMI of 30 kg/m^2^ or greater
			Hyperthyroidism	Pediatrics: depression, dizziness, arthralgia, pyrexia, influenza, and ligament sprain	
			Taking or within 14 days of stopping monoamine oxidase inhibitors		
Naltrexone and bupropion (CONTRAVE) [139]	Naltrexone, an opioid antagonist, and bupropion, an aminoketone antidepressant	2014	Seizure disorders	Nausea, vomiting, constipation, diarrhea, dry mouth, headache, dizziness, and insomnia	Indicated for patients with a BMI of 27 kg/m^2^ or more with at least one weight-related ailment or a BMI of 30 kg/m^2^ or greater
			Use of benzodiazepines, barbiturates, and antiepileptic drugs		Effect on cardiovascular morbidity and
			Chronic opioid use		mortality not established.
			During or within 14 days of taking monoamine oxidase inhibitors (MAOI)		Safety and effectiveness in combination with other weight loss products not established
Lorcaserin hydrochloride (BELVIQ) [140]	A serotonin 2C receptor agonist	2012	Pregnancy	Headache, dizziness, fatigue, nausea, dry mouth, and constipation, and hypoglycemia in diabetic patients	Indicated for patients with a BMI of 27 kg/m^2^ or more with at least one weight-related ailment or a BMI of 30 kg/m^2^ or greater
					Effect on cardiovascular morbidity and
					mortality not established.
Orlistat (XENICAL) [141]	Reversible inhibitor of gastrointestinal lipases	1999	Pregnancy	Oily spotting, flatus with discharge, fecal urgency fatty/oily stool, oily evacuation, increased defecation, and fecal incontinence.	Also indicated to reduce the risk for weight regain after prior weight loss.
			Chronic malabsorption syndrome		Gastrointestinal events may increase when taken with a
			Cholestasis		diet high in fat (>30% total daily calories from fat).
					Patients encouraged to take fat-soluble vitamins to
					ensure adequate nutrition.
Phentermine hydrochloride (SUPRENZATM) [142]	Sympathomimetic amine anorectic	1959	History of cardiovascular disease	Palpitation, tachycardia, elevation of blood pressure, ischemic events, restlessness, dizziness, insomnia, euphoria, dysphoria, tremor, headache, psychosis, dryness of the mouth, unpleasant taste, diarrhea, constipation, Impotence, changes in libido.	Indicated for patients with a BMI of 27 kg/m^2^ or more with at least one weight-related ailment or a BMI of 30 kg/m^2^ or greater
			During or within 14 days following the administration of monoamine oxidase inhibitors		Coadministration with other drugs for weight loss not recommended (safety and efficacy not established)
			Hyperthyroidism		
			Glaucoma		
			History of drug abuse		
			Pregnancy		
			Nursing		
			Known hypersensitivity, or idiosyncrasy to the sympathomimetic amines		

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
