# Peer review of "Hypertension Related to Obesity: Pathogenesis, Characteristics and Factors for Control"

_ijms, 2022, doi:10.3390/ijms232012305_

Round 1

Reviewer 1 Report

Dear Authors,

 The manuscript presents an important topic regarding hypertension-related obesity. The authors presented factors and mechanisms associated with obesity and hypertension in the form of a literature review.

I have some minor considerations regarding missing elements in the manuscript.

  1. Please explain the meaning of the rA correlation coefficient in the text (line 125)
  2. Gene symbols should be italicized.
  3. There are several shortcuts without explanation in the manuscript (e.g., full gene names). Please also explain the shortcuts in the proper order in the text.
  4. Lines (147-149): please, briefly explain in the text what the authors exactly mean by" false positivity"?
  5. Line 206: mRNA expression of what particular gene?
  6. Line 207: please put the name of a mentioned online database
  7. Line 237: Please briefly explain the fision term to readers.
  8. Line 191: Do the authors mean "the role of histone modification"?
  9. Line 337 ": "not necessary
  10. Line 373: Natriuresis should be lowercased
  11. Line 378-380: Is detailed data about the new drug (name in the market and name of the company) relevant to mention in the manuscript? Are there data suggesting its promising role in managing obesity-related hypertension? Thus, please briefly explain the unique properties of a new drug that differentiate them from similar drugs (e.g., mechanism of action, the structure of the active substance, the drug form?)
  12. Line 362: citing Table 1 seems to be unnecessary there.
  13. Line 442: Il-20 word is doubled.
  14. The line from 429-... The manuscript's "Experimental pathways" section mainly concerns obesity alone. Please, if possible, add some information about the action of described factors in hypertension in obesity.
  15. The conclusion section should underline the knowledge presented in the manuscript. Please explain why in conclusion, only the most recent and potent Tirzepatide offers new hopes for patients with hypertension-related obesity.

Reviewer 2 Report

The manuscript under review is devoted to an urgent topic - the relationship between obesity and arterial hypertension. The authors considered in details all aspects of this problem. Particularly very interesting and new in content is the section on epigenetic factors influencing the development of hypertension in the obese patients. In general, the article presents a fairly complete and well-written overview that helps to understand such a complex problem. The publication of this article will be useful for a wide range of readers in the field of medicine.

Reviewer 3 Report

This review article provides an overview of the pathogenesis and treatment of hypertension associated with obesity. While this review paper provides up-to-date information on obesity, this reviewer has several concerns as listed below:

1.      While the aim of this review paper is to gain insight on the pathogenesis and effective treatment strategies for hypertension associate with obesity, information on blood pressure is sparse throughout the manuscript. In particular, in the section 5.2. Experimental Pathways, little information is provided on the effects of these specific pathways on blood pressure. The authors should provide literature evidence that reported blood pressure lowering effects of these experimental pathways in obese animals and humans. In its present form, this reviewer feels that this review paper is just an educational review on obesity.

2.      The section 2. Pathogenesis of hypertension associated with obesity doesn't provide sufficient information. The authors should expand and provide more information. For example, changes in the generation of endothelium-derived constricting as well as relaxing factors, impaired activity of vascular potassium channels, increased adipocyte-secreted aldosterone levels, dyslipidemia etc. all of which could increase blood pressure during obesity.  

3.      The table is hard to read. This reviewer thinks placing the table horizontally will make it easier to read.

4.      Some paragraphs are too long to read. These paragraphs should be set off by appropriate paragraph breaks. This would increase the readability of the text.

Round 2

Reviewer 3 Report

The authors have adequately responded to all my concerns. The manuscript is now greatly improved and the revised manuscript is recommended for consideration for publication.